# Analysis of the Thermal Behavior of a New Structure of Protected Agriculture Established in a Region of Tropical Climate Conditions

Edwin Villagrán * and Andrea Rodriguez

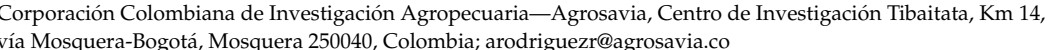

Corporación Colombiana de Investigación Agropecuaria—Agrosavia, Centro de Investigación Tibaitata, Km 14, vía Mosquera-Bogotá, Mosquera 250040, Colombia; arodriguezr@agrosavia.co
* Correspondence: edwina.villagranm@utadeo.edu.co; Tel.: +57-1-4227-300 (ext. 1239)

**Abstract:** Determining airflow patterns and their effect on the distribution of microclimate variables such as temperature is one of the most important activities in naturally ventilated protected agricultural structures. In tropical countries, this information is used by farmers and decision makers when defining climate management strategies and for crop-specific cultural work. The objective of this research was to implement a validated Computational Fluid Dynamics (CFD) model in 3D to determine the aerodynamic and thermal behavior of a new protected agricultural structure established in a warm climate region in the Dominican Republic. The numerical evaluation of the structure was carried out for the hours of the daytime period (6–17 h), the results found allowed to define that the CFD model generates satisfactory predictions of the variables evaluated. Additionally, it was found that airflow patterns are strongly affected by the presence of porous insect screens, which generate moderate velocity flows ($<0.73$ m s$^{-1}$) inside the structure. It was also identified that the value of the average temperature inside the structure is directly related to the air flows, the level of radiation and the temperature of the outside environment.

**Keywords:** protected agriculture; microclimate; CFD model; simulation; tropical

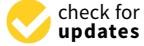



## 1. Introduction

In tropical countries with hot climate conditions, protected agriculture based on passive climate control structures is still in constant search and adaptation to appropriate technologies that allow an increase in production [1]. The main problem of the current designs of structures implemented in these countries is related to the generation of inadequate microclimatic conditions for the growth and development of the plants [2,3]. The most important microclimatic limitation is associated with the high temperature values generated inside the structures, which produces thermal stress conditions on the crops [4–6].

In tropical regions the negative effects of climate change are increasingly frequent and severe [7]. In this context, it is to be expected that in general, this region will have more restrictive climatic conditions for agriculture focused on open field production in the future [8,9]. In this scenario, agriculture will be exposed to different biotic and abiotic factors and constraints that will affect crop yields and thus the sustainability of food production systems [10].

On the other hand, current population growth is coupled with the need to ensure food security of nations [11]. They constantly demand an increase in crop yields from the agricultural sector in charge of food production, with a particular slogan and wanting the highest production with a decreasing use of natural resources [12,13]. There is protected agriculture with a low technological level, but with designs are suitable for the climatic conditions of each region. It can be an interesting alternative of sustainable intensification for small and medium producers who do not have the economic resources to acquire high-tech greenhouses [14,15].

The management of the microclimate generated inside a protected passive agricultural structure is carried out through the phenomenon of natural ventilation. This method of climate control is widely used in various countries around the world and in various structures in each region [9,16]. The air flows generated by natural ventilation are responsible for regulating the excess temperature and humidity inside the structure, and in turn, this is the only source of carbon enrichment of the internal atmosphere of the structure [17]. In the same way, natural ventilation together with solar radiation are the main variables that affect the spatial distribution of the microclimate inside a protected agricultural structure [18].

The study of the thermal behavior of a protected agricultural structure is a fundamental task to carry. From this arises important information that, in the future, the producer will use for the management of the microclimate and for the tasks of agronomic management of the established crops. One of the most approached and used methodologies for this type of study is numerical CFD simulation, mainly because of its comparative advantages over other experimental methodologies [19–22]. CFD simulation allows to obtain the spatial distribution fields of air flows and temperature inside a naturally ventilated structure, such as roof structures used in agriculture [23,24].

The main objectives of this research were (i) to evaluate and validate a three-dimensional CFD numerical simulation model applicable to a naturally ventilated protected agriculture structure and (ii) to evaluate through numerical simulation using the validated model, the aerodynamic behavior of the air flow patterns and the spatial distribution of the temperature inside a new protected agriculture structure built in a low latitude region in the Dominican Republic. The evaluation and validation of the numerical model was carried out for the daytime hours between 7:00 a.m. and 5:00 p.m.

## 2. Materials and Methods

### 2.1. Description of the Structure and Experimental Arrangement

In this research, a new protected agricultural structure of 560 m$^2$ of covered area was evaluated, designed for the dominant climatic conditions of the province of La Vega in the Dominican Republic. The overall dimensions of the structure can be seen in Figure 1. This new structure was covered on both sides only with porous insect proof mesh, areas that would function as a screenhouse structure. While the central area was covered with a commercial greenhouse polyethylene film, this central area was equipped with two roof ventilation areas, ventilation areas that were equipped with insect-proof porous screen.

For the validation of the numerical model, an experimental trial was carried out that included the collection and recording of climatic data inside and outside the structure. This registration was carried out during a total of 45 days with a frequency of ten minutes. The collection of experimental data included the recording of outdoor variables such as temperature (°C), solar radiation (W m$^{-2}$), speed (m s$^{-1}$) and wind direction. While inside the structure, 5 temperature recorders (test points) were distributed at a height of 1.6 m above ground level (Figure 2).

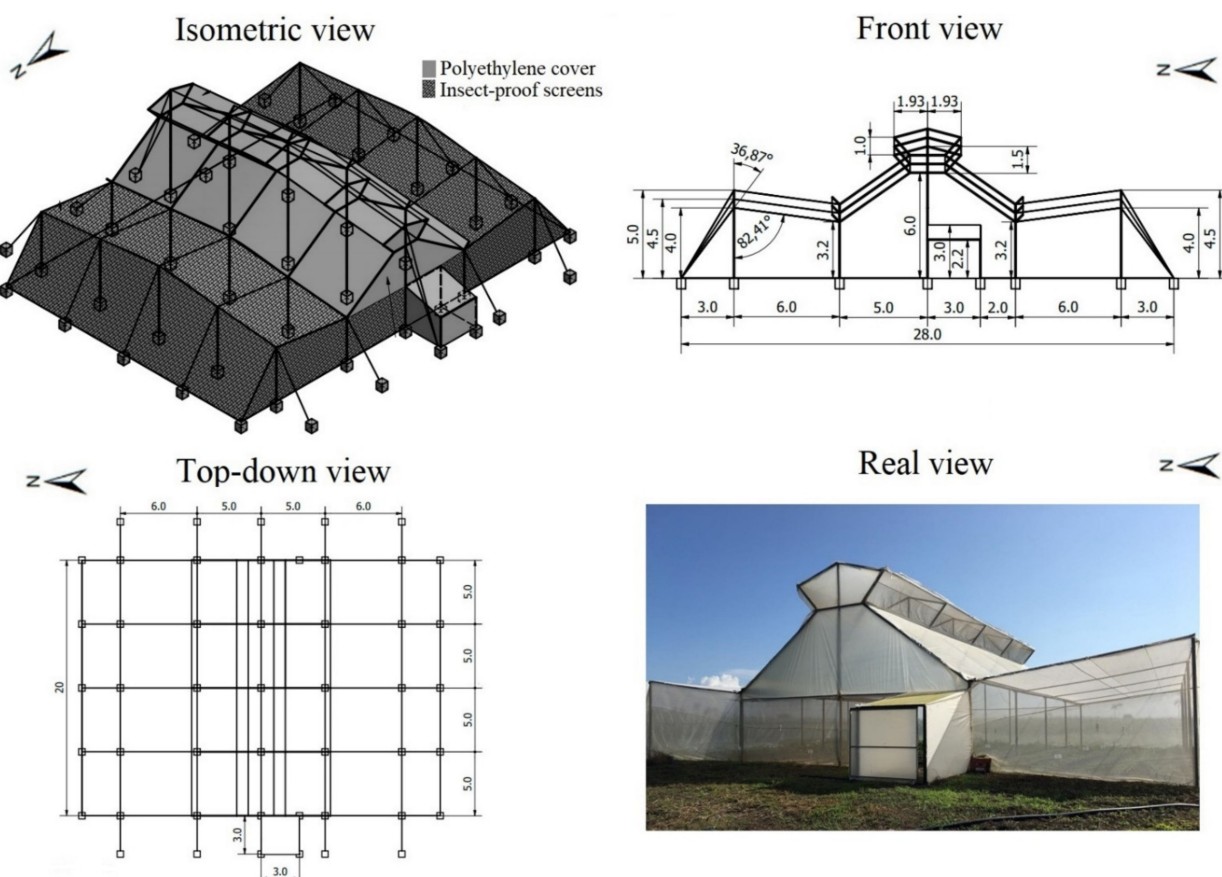

**Figure 1.** Overall dimensions and actual view of the evaluated structure.

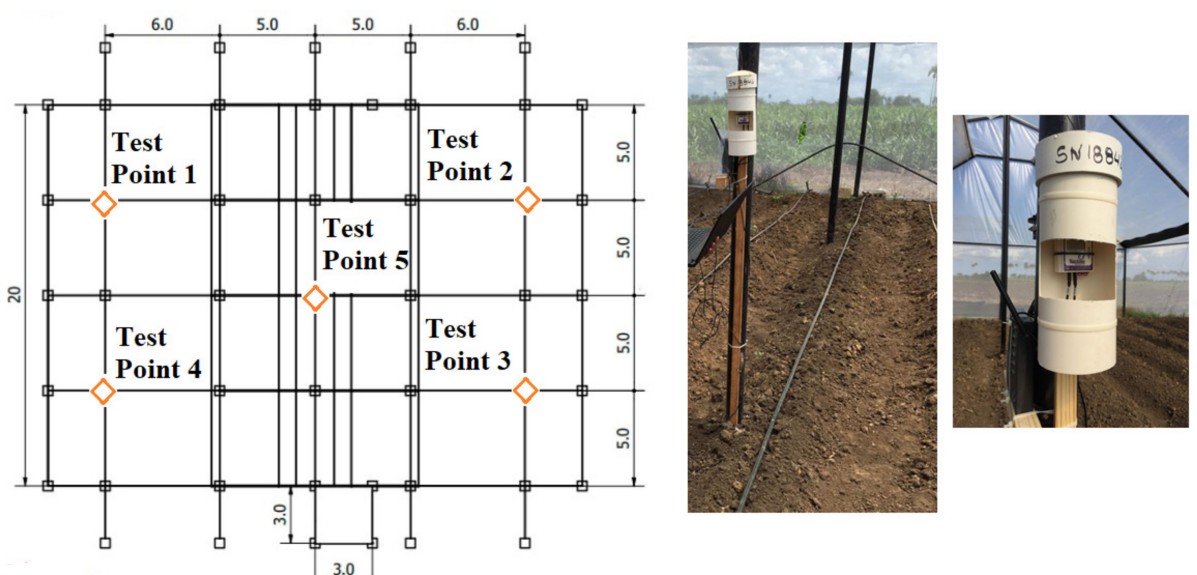

**Figure 2.** Schematic of the experimental climate data record.

### 2.2. Physical and Mathematical Model and Boundary Conditions

The CFD methodology allows the calculation of air flow patterns and heat distribution patterns generated inside a protected agricultural structure. The numerical CFD simulation is divided into 3 main stages, pre-process, solution and post-process. In the pre-process the physical problem to be solved is defined, the virtual model of the geometry of the structure object of study is generated, the size of the computational domain is defined and

the size of the numerical mesh for the whole computational domain. In the solution, the numerical models are selected for simulation, the boundary and convergence conditions are defined and the initial simulation conditions are established. Finally, in the post-process, the exploration and obtaining of qualitative and quantitative data necessary to validate the CFD model and to carry out the analysis corresponding to the objective of the investigation is carried out.

For the pre-processing phase, a large, coupled computer domain was built (Figure 3). Through this computational domain, the modeling and numerical simulation of the natural ventilation phenomenon inside the analyzed protected agricultural structure is generated [25,26]. The size of the computational domain must be defined in order to allow an adequate and realistic development of the airflow, thus obtaining an adequate prediction of the microclimate behavior [9,15]. For this research it was defined that the minimum distance from the edges of the computational domain to each side of the structure was 20 H and the height of the computational domain 10 H, where H is the maximum height of the structure, dimensions that are similar to those defined in other studies of natural ventilation in greenhouses [9,27].

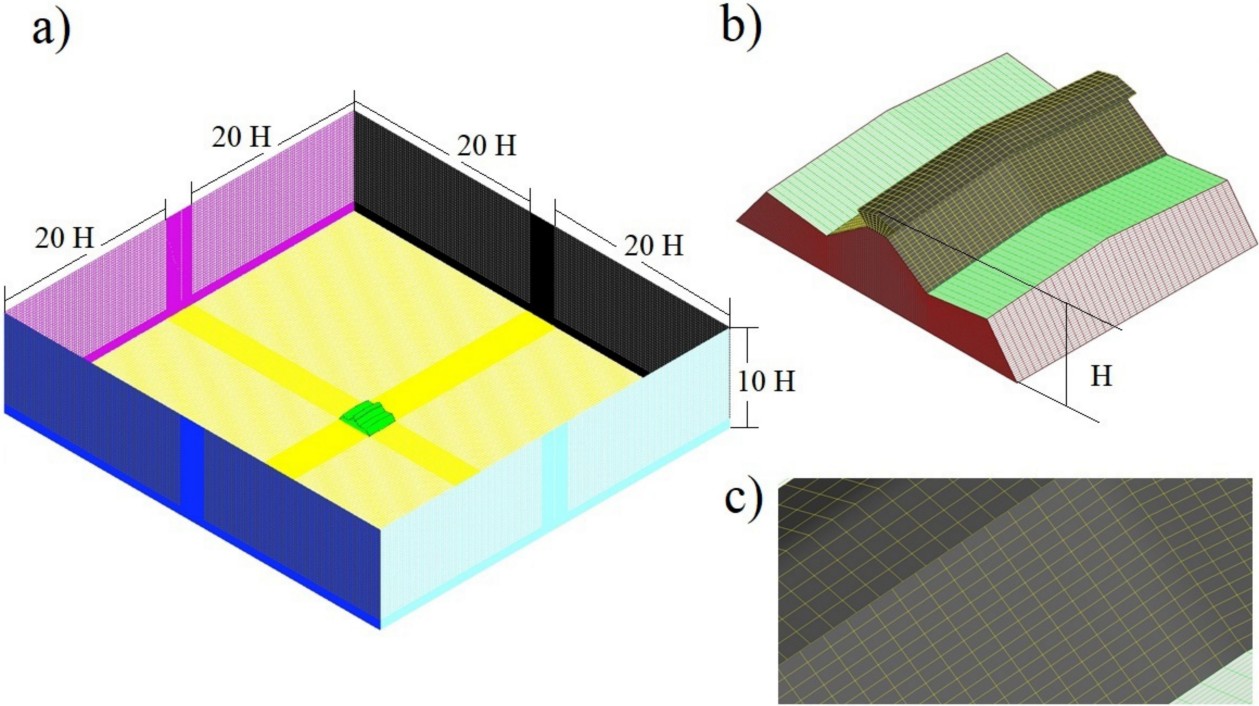

**Figure 3.** (**a**) size of the computational domain, (**b**) grid of the structure and (**c**) detail of the numerical grid of the structure.

The numerical mesh of the computational domain and the structure was defined from a sensitivity analysis where eleven numerical grids of different size were evaluated, this process included a refinement on the areas where the biggest thermal gradients are produced such as the floor, the walls and the cover of the structure [28,29]. The graphic details of the mesh can be seen in Figure 3.

The results obtained from this sensitivity analysis will be discussed later in this document in the results section. On the other hand, once the size of the numerical grid was defined, we proceeded to evaluate the orthogonal quality of the grid elements, obtaining a value of 0.96, which is considered within the range of excellent quality [30–32]. Checking the quality parameters of the grid is an indispensable step in numerical simulation studies, since this factor is one of the most influential on the quality of CFD simulation model predictions [33,34].

To do this, computational domain are coupled with the general conservation equations for energy (Equation (1)), momentum (Equation (2)) and mass (Equation (3)). This set of

equations is known as the Navier–Stokes equations and allows modeling and simulating the flow of a fluid and its relationship with the transfer of heat and mass inside a protected agricultural structure.

$$\nabla(\vec{v}(\rho E + P)) = \nabla(k_{eff}\nabla T - \sum_j h_j \vec{J_j} + ((\bar{\bar{\tau}}_{eff})(\vec{v}))) \tag{1}$$

$$\nabla(\rho \vec{v}\vec{v}) = -\nabla P + \nabla(\bar{\bar{\tau}}) + \rho \vec{g} \tag{2}$$

$$\nabla(\rho \vec{v}) = 0 \tag{3}$$

where $\vec{v}$ is the velocity vector, $\rho$ is the density of the fluid under study, $\vec{g}$ is the gravitational force, $\bar{\bar{\tau}}$ is Reynolds' stress tensor, $P$ is the pressure, $T$ represents the temperature, $E$ is the energy of the flow, $k_{eff}$ the effective conductivity, $h_j$ is the enthalpy and $\bar{\bar{\tau}}_{eff}$ and $\vec{J_j}$ are the viscosity shear and the species diffusion flow, respectively.

The soil of the structure and the computational domain, as well as the polyethylene cover were modeled with wall contour conditions, imposing on them the properties summarized in Table 1. The boundaries of the computational domain parallel to the direction of the outside air flow were modeled with a symmetrical property boundary condition. While the inlet and outlet airflow limits were modeled with inlet velocity and outlet pressure boundary conditions, respectively.

**Table 1.** Thermophysical and optical properties of the materials used in the simulation model. Adapted from Li et al. [35].

| Property | Air | Agricultural Soil | Polyethylene Cover |
|---|---|---|---|
| Density ($\rho$, kg m$^{-3}$) | 1.225 | 1700 | 950 |
| Thermal conductivity (k, W m$^{-1}$ K$^{-1}$) | 0.0242 | 0.85 | 0.19 |
| Specific heat (Cp, J K$^{-1}$ kg$^{-1}$) | 1006.43 | 1010 | 1600 |
| Absorptivity coefficient | 0.10 | 0.5 | 0.15 |
| Refractive index | 1.0 | 1.92 | 1.7 |
| Emissivity | 0.86 | 0.90 | 0.85 |

At the air entry limit, a logarithmic air flow inlet profile was imposed by a user-defined function, according to the specific local terrain conditions and in accordance by Villagrán et al. [15]. A model of the inlet airflow turbulence was also coupled to this airflow, in this case the standard *k-ε* model was considered, a model widely used and successfully validated in this type of studies [36,37]. This model is represented by one equation for turbulent kinetic energy (Equation (4)) and another for dissipation rate (Equation (5)) [28].

$$\frac{\partial}{\partial x}(\rho k) = \frac{\partial}{\partial x_j}\left[\left(\mu + \frac{\partial k}{x_j}\right)\frac{\partial k}{\partial x_j}\right] + G_k + G_b - \rho\epsilon - Y_M \tag{4}$$

$$\frac{\partial}{\partial t}(\rho\varepsilon) = \frac{\partial}{\partial x_i}\left[\left(\mu + \frac{\mu_t}{\sigma}\right)\frac{\partial \epsilon}{\partial x_i}\right] + \rho C_1 S_\epsilon - \rho C_2 \frac{\epsilon^2}{k + \sqrt{v\epsilon}} + C_{1\epsilon}\frac{\epsilon}{k}C_{3\epsilon}G_b k \tag{5}$$

where $\mu$ and $\mu_t$ are the viscosity and the turbulent viscosity of the fluid and $G_b$ and $G_k$ represent the generation of turbulent kinetic energy due to buoyancy and speed, respectively. $Y_M$ is the fluctuating expansion in turbulence due to the overall dissipation rate, $\sigma_k$ and $\sigma_\epsilon$ are Prandtl's turbulent numbers for $k$ and $\varepsilon$, $v$ is the coefficient of kinematic viscosity and $C_{1\epsilon}$, $C_{2\epsilon}$, $C_\mu$, $\sigma_k$, $\sigma_\epsilon$ are constant with experimentally determined values [38].

The presence of insect-proof porous screens was simulated by activating the porous media type boundary condition in the model. To complete this, the aerodynamic parameters for a 16.1 by 10.2 thread per cm$^{-2}$ porous screen must be entered into the model. The aerodynamic parameters used in this research are those successfully implemented in the

study developed by Flores Velasquez et al. [39]. The flow of air through the porous screens was simulated by adopting the Darcy–Forchheimer's law (Equation (6)).

$$\frac{\partial p}{\partial x} = \frac{\mu}{K}u + \rho\frac{Y}{\sqrt{K}}u|u| \tag{6}$$

where $Y$ and $K$ represent the loss of non-linear momentum and the permeability of the porous medium, these factors are determined from aerodynamic equations obtained in experimental tests conducted in wind tunnel [40]. On the other hand, $u$ is the air velocity, $\mu$ is the dynamic viscosity of the fluid and $\rho$ and $\partial x$ are the density of the air and the thickness of the porous medium, respectively.

Finally, in the upper part of the computational domain, a boundary was established for the solar radiation condition that will be modeled by coupling the discrete order (DO) model. This model is widely used to solve the radiative transfer equation in studies of buildings or constructions that have a semi-transparent roof, such as screenhouses and greenhouses [41]. The model is described by Equation (7).

$$\nabla.\left(I_\lambda\left(\vec{r},\vec{s}\right)\vec{s}\right) + (a_\lambda + \sigma_s)I_\lambda\left(\vec{r},\vec{s}\right) = a_\lambda n^2\frac{\sigma T^4}{\pi} + \frac{\sigma_s}{4\pi}\int_0^{4\pi}I_\lambda\left(\vec{r},\vec{s}\prime\right)\Phi\left(\vec{s}.\vec{s}\prime\right)d\Omega\prime \tag{7}$$

where $I_\lambda$ is the intensity of radiation at a wavelength, $\vec{r}$, $\vec{s}$ are the vectors that indicate the position and direction, respectively, $\vec{s}\prime$ is the direction vector of the scatter, $\sigma_s$, $a_\lambda$ are the coefficients of dispersion and spectral absorption, $n$ is the refractive index, $\nabla$ is the divergence operator, $\sigma$ is Stefan-Boltzmann's constant and $\Phi$, $T$ and $\Omega$ are the phase function, the local temperature (°C) and the solid angle, respectively. Likewise, the chimney effect of natural ventilation produced by the effect of air buoyancy was simulated by using the Boussinesq approximation in the model [42].

For the process phase it was established to execute numerical simulations under steady state conditions, by using the SIMPLE algorithm, also using a second order spatial discretization scheme. The residuals established for the momentum, continuity, turbulence and radiation equations were $10^{-3}$ while for the energy equation it was established at $10^{-6}$. The numerical simulations were run on a high-performance computer equipped with an Intel® Xeon W-2155 processor with twenty cores at 3.30 GHz and a RAM capacity of 64 GB.

### 2.3. Developed Simulation Scenarios

For the development of the simulations and to simplify the numerical process and its solution time, no type of culture was included in the computational domain that would contribute as a source term to the moment and energy equations. The simulated scenarios corresponded to the average climatic conditions obtained for the hours of the day (6–17 h) during the period of the experimental trial, these conditions are summarized in Table 2.

**Table 2.** Mean hourly weather conditions used as input parameters to the CFD model.

| Hour | Air Temperature [°C] | Solar Radiation [W m$^{-2}$] | Wind Velocity [m s$^{-1}$] | Wind Direction |
|---|---|---|---|---|
| Hour 06 | 21.3 | 30.1 | 0.2 | ESE |
| Hour 07 | 21.8 | 66.1 | 0.3 | ESE |
| Hour 08 | 23.6 | 233.2 | 0.5 | S |
| Hour 09 | 25.6 | 448.2 | 0.7 | S |
| Hour 10 | 27.3 | 643.5 | 1.3 | S |
| Hour 11 | 28.7 | 793.9 | 1.9 | S |
| Hour 12 | 29.8 | 873.7 | 2.5 | S |
| Hour 13 | 30.5 | 859.7 | 3.1 | S |
| Hour 14 | 30.8 | 860.2 | 3.1 | ESE |
| Hour 15 | 30.8 | 732.5 | 3.3 | S |
| Hour 16 | 30.4 | 433.8 | 3.5 | ESE |
| Hour 17 | 29.4 | 211.6 | 3.3 | ESE |

*2.4. Model Validation*

To determine the validity of the numerical model and the quality of the prediction of the microclimate generated inside the structure, the comparison of data obtained in a numerical and experimental way was carried out. The real average temperature obtained for each test points (1–5) in each scenario evaluated during the experimental trial was compared with the temperature obtained through the CFD model under the simulation conditions mentioned in Table 2.

This comparison was made under a qualitative approach by constructing the trend graph of the measured and simulated temperature data. Complementary analysis with a quantitative evaluation by comparing these data sets, using goodness-of-fit parameters such as the Root Mean Square Error (RMSE) and the Mean Absolute Percentage Error (MAPE), Equations (8) and (9), respectively.

$$RMSE = \sqrt{\sum_{i=1}^{m} \frac{(Tmi - Tsi)^2}{m}} \tag{8}$$

$$MAPE = \frac{1}{m} \sum_{i=1}^{m} \left| \frac{Tmi - Tsi}{Tmi} \times 100 \right| \tag{9}$$

where *m* is the number of data sampled, *Tmi* and *Tsi* are the temperature values for a specific moment measured and simulated, respectively.

## 3. Results

*3.1. Test of Independence of Numerical Grid*

The selected numerical grid had a size of 11,075,310 elements formed in an unstructured grid (Grid number 7). This size was determined once the independence test was performed for a total of eleven grids that varied in their number of numerical elements between (446,326–18,753,108). For this analysis, a steady state simulation was executed for each of the numerical grids evaluated keeping constant the input conditions established for the hour 13 and that can be consulted in Table 2.

After convergence of the simulations is reached, the analysis focuses on observing the variation of average values of temperature and air speed inside the evaluated structure (Figure 4). For this specific case it can be observed that the numerical solution for both temperature and wind velocity do not show a greater variation after grid number 7. Therefore, this is the size of the numerical grid that allows obtaining numerical solutions independent of the size of the numerical grid and with a moderate computational cost, as it has already been reported by studies such as the one developed by Villagrán et al. [9] and Akrami et al. [43].

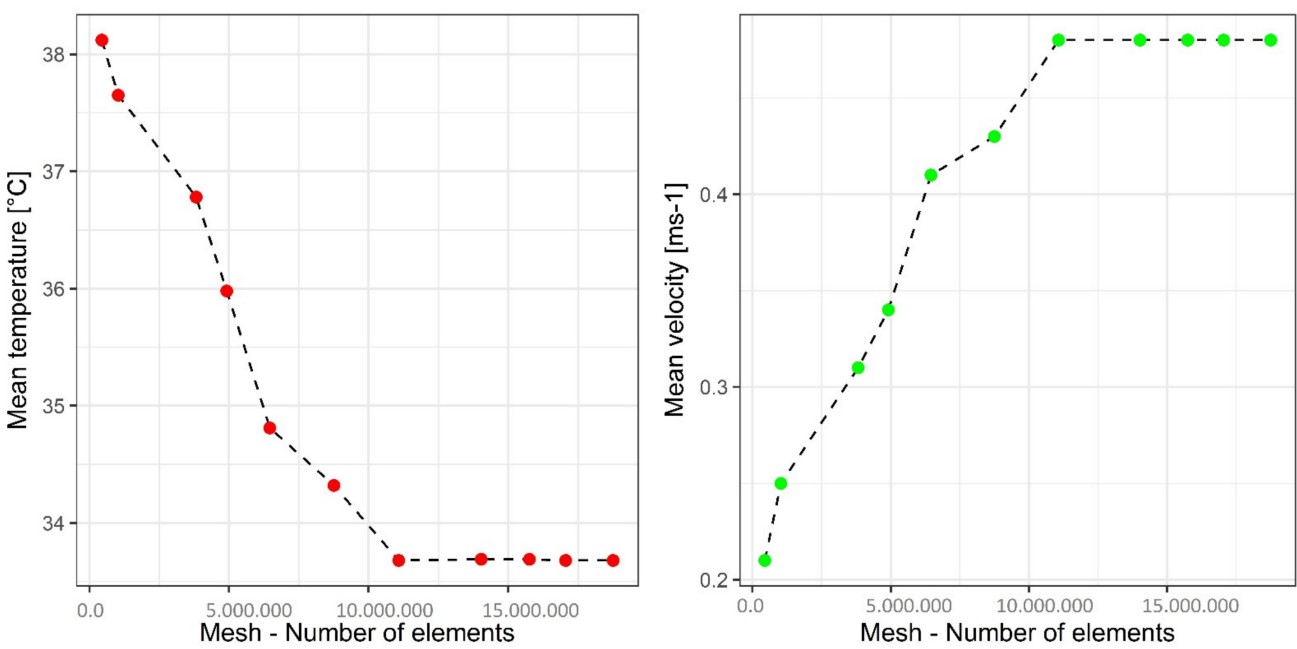

**Figure 4.** Results of the numerical grid independence test.

### 3.2. Numerical Model Validation

The qualitative results of the temporal trends on an hourly scale of the average temperatures obtained through simulation and experimentation at each of the sampling points (1–5) can be seen in Figure 5. In general, the data sets show very similar trend and magnitude behavior during the evaluated time scale (6–17 h). Therefore, this simple comparison allows to define that the numerical model is performing an adequate prediction of the thermal behavior of the evaluated structure.

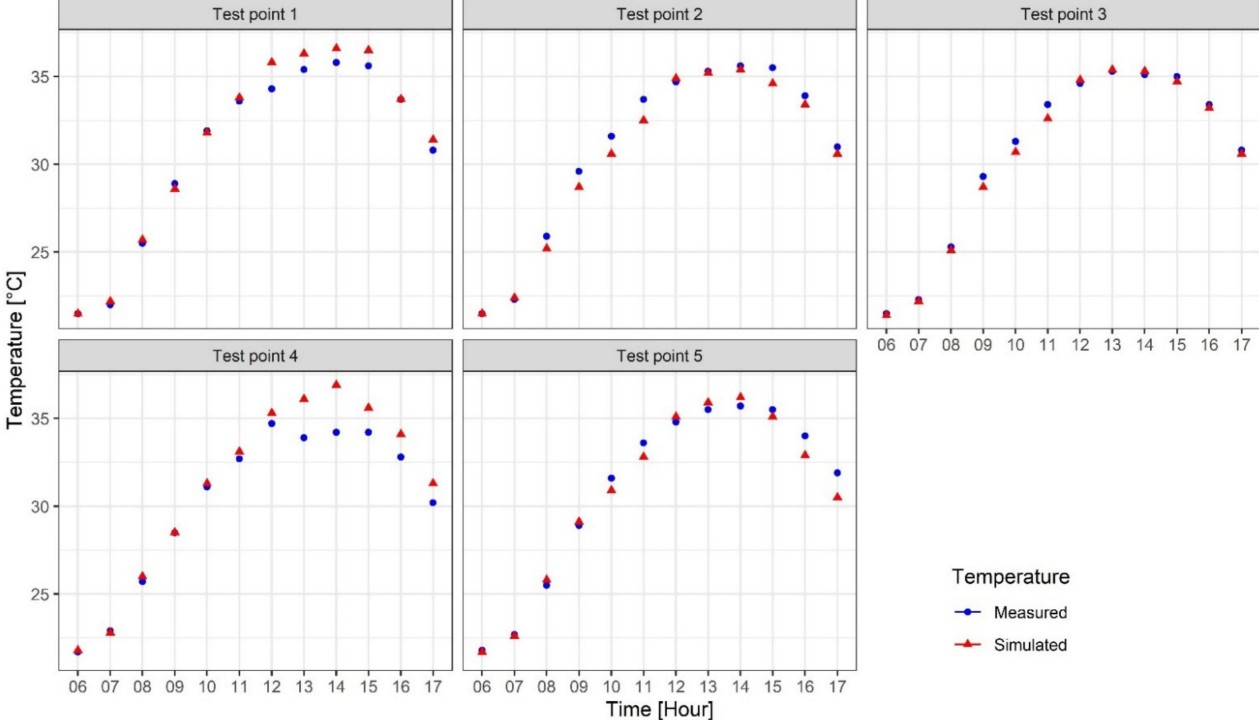

**Figure 5.** Temporal behavior of temperature data simulated and measured at each test point.

The qualitative validation was complemented by a quantitative evaluation through goodness-of-fit parameters commonly used in this type of study. The results obtained at each measurement point for RMSE and MAPE can be found in Table 3. The RMSE for the temperature varies between a minimum value of 0.75 °C for measuring point 3 and a maximum of 2.94 °C for measuring point 4, values that are similar to those reported in a numerical study of natural ventilation developed by Villagrán and Bojacá [28]. On the other hand, ASM results ranged between 0.96% and 2.64%, values that, being lower than 10%, allow us to conclude that the model has a highly accurate forecasting capacity, as reported by Montaño et al. [44].

**Table 3.** Numerical results of the validation process of the CFD model.

| Test Point | RMSE [°C] | MAPE [%] |
|---|---|---|
| Test point 1 | 1.41 | 1.43 |
| Test point 2 | 1.61 | 1.63 |
| Test point 3 | 0.75 | 0.96 |
| Test point 4 | 2.94 | 2.64 |
| Test point 5 | 0.83 | 1.62 |

*3.3. Air Flow Patterns*

Be seen in Figure 6. Inside the structure the movement of the air velocity vectors present a generalized behavior where the air flow enters the structure through the south side wall and the region of the roof that is formed by porous mesh. These flows then pass through the structure until they leave the interior of the structure through the north side sidewall and the roof region covered by porous screen (Figure 6). This type of behavior is very similar to that reported in screen house structures such as those analyzed in the studies developed by Villagrán and Jaramillo [5] and Flores Velasquez et al. [45]. Studies that have as similarity the aerodynamic analysis of structures of screen houses with asymmetric roofs, like the structure of this research.

A differentiated behavior was observed under the simulated scenario for hour 07 (Figure 6). In this case it is observed that the air flows show recirculation patterns from the central zone of the greenhouse that are directed to the sides and to the facades of the structure. This behavior in this case happens because in these early morning hours the wind speed from the outside is low (<0.3 m s$^{-1}$). Therefore the movement of air inside the structure is strongly influenced by the free convection through air buoyancy, due to the effect of the heat generated by the solar radiation entering from the outside [15,46,47].

Another of the behaviors observed in the distributions of the air flow inside the structure, is the effect generated due to the presence of the polyethylene sheet located on the central zone of the structure, zone that serves as a passive greenhouse. The presence of this plastic film generates on the one hand a suction effect towards the interior of the structure, which promotes a greater movement of air towards the region where the crops will be developed (Figure 6). On the other hand, a recirculating loop is also generated in the upper zone between the region of the gutters and the roof ventilation areas arranged in the structure. This region presents an air exchange between the indoor and outdoor environments due to the movement generated by the effect of the buoyancy of the warm air of the indoor environment and by the wind effect produced by the high speed conditions (>2 m/s) in the outdoor environment [6,48–51].

For the quantitative evaluation of the air velocity inside the structure, a total of 96,387 data were extracted from the inside volume. For these data sets, the average velocity ($V_m$) and the normalized velocity ($U_0$) were calculated, which represents the relationship between the indoor air speed and the outdoor air velocity (Table 4). The values for $V_m$ oscillated between a minimum of $0.14 \pm 0.11$ m s$^{-1}$ for hour 06 and a maximum of $0.72 \pm 0.23$ m s$^{-1}$ for hour 16, values that coincide with the time when the minimum and maximum values of wind speed in the external environment are presented.

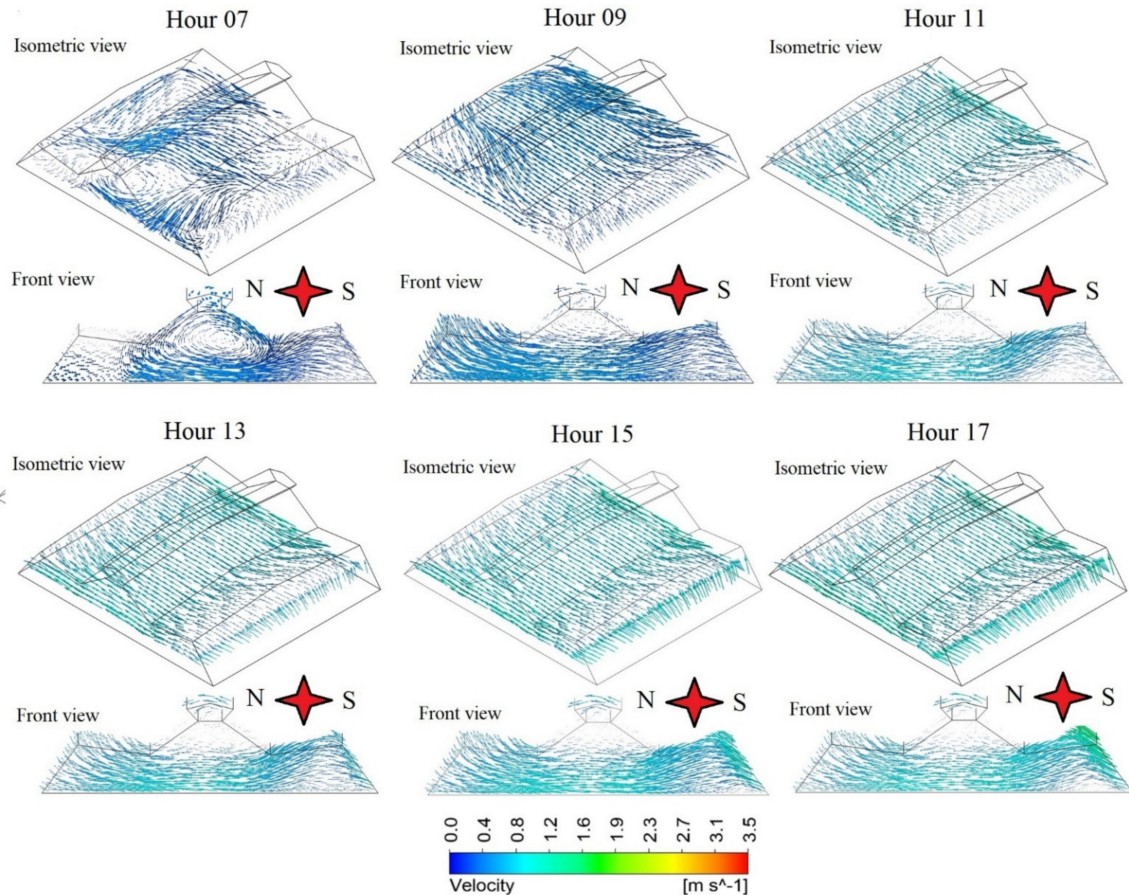

**Figure 6.** Simulated airflow patterns (m s$^{-1}$) for each scenario evaluated.

**Table 4.** Numerical parameters obtained for air flow patterns.

| Hour | $V_m$ [m s$^{-1}$] | $U_0$ [%] | Hour | $V_m$ [m s$^{-1}$] | $U_0$ [%] |
|---|---|---|---|---|---|
| Hour 06 | $0.14 \pm 0.11$ | $70.3 \pm 17.1$ | Hour 12 | $0.48 \pm 0.22$ | $19.2 \pm 9.4$ |
| Hour 07 | $0.19 \pm 0.08$ | $65.8 \pm 29.6$ | Hour 13 | $0.57 \pm 0.21$ | $18.3 \pm 7.1$ |
| Hour 08 | $0.21 \pm 0.08$ | $42.9 \pm 16.3$ | Hour 14 | $0.58 \pm 0.20$ | $18.5 \pm 6.9$ |
| Hour 09 | $0.31 \pm 0.10$ | $45.6 \pm 14.6$ | Hour 15 | $0.64 \pm 0.22$ | $19.6 \pm 6.6$ |
| Hour 10 | $0.51 \pm 0.19$ | $39.8 \pm 14.7$ | Hour 16 | $0.72 \pm 0.23$ | $20.6 \pm 6.8$ |
| Hour 11 | $0.52 \pm 0.21$ | $27.5 \pm 11.4$ | Hour 17 | $0.71 \pm 0.24$ | $21.6 \pm 7.6$ |

Likewise, it can be seen that the $V_m$ is gradually increasing as the wind speed increases outside. These mean velocity values obtained in this research coincide with those reported in numerical studies of mesh house structures such as the one performed by Flores Velasquez and Montero [52] and with experimental studies developed with sonic anemometry as the one performed by Teitel et al. [53].

Finally, the $U_0$ values allow to identify the speed reduction of the air flows generated due to the presence of the insect-proof porous screen and its negative effect on the loss of impulse of those air flows [54,55]. The values of $U_0$ ranged from a minimum of 18.3% for hour 13 to a maximum of 70.3% for hour 06, values that allow us to conclude that the loss of airflow speed ranges from 29.7% to 81.7%.

Likewise, it can be observed that as the air speed increases outside, the loss of speed of the air flows inside the structure is greater, this has been previously reported by Teitel et al. [53]. This is because with a higher outside wind speed, the pressure drop of the airflow over a porous screen increases significantly [56].

### 3.4. Spatial Temperature Distribution

The distribution of the simulated temperature in a plan view section at a height of 1.5 m above ground level. Just as in a cross section over the central area of the evaluated structure can be seen in Figure 7. In general terms it can be identified that the thermal spatial distributions show a behavior where the region with the lowest temperature value is located just on the south side wall. Region where air flows enter from the outside environment, this is because in naturally ventilated structures, the thermal distribution has a direct relationship with the movement of air flows [32,57,58].

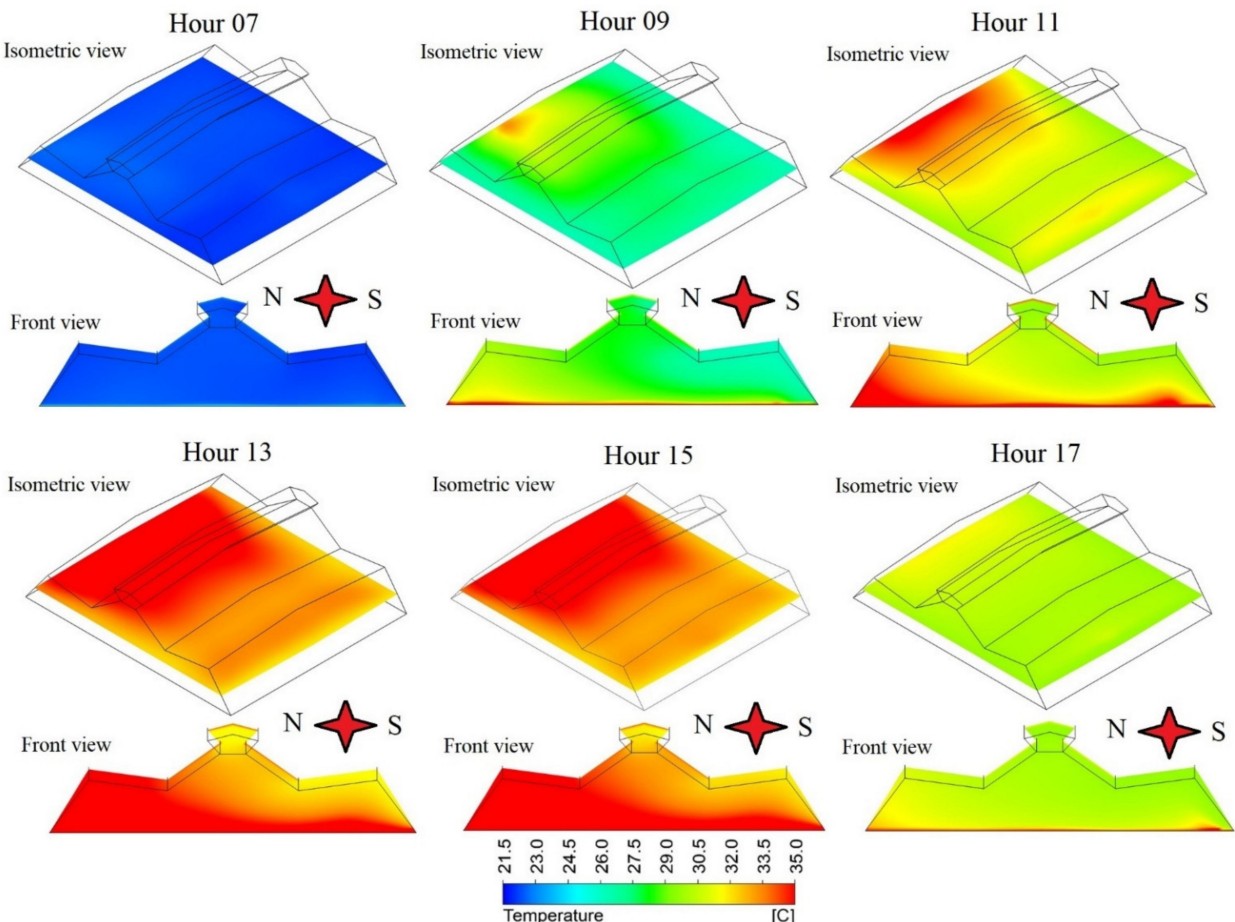

**Figure 7.** Fields of thermal distribution (°C) obtained for each evaluated scenario.

It is also observed that the air flows in its displacement pattern generate an energetic drag towards the side wall of the north side, which generates a region of higher temperature over this area. This energy drag occurs from the heat transfer phenomena that occur between the interior air, the soil and the roof of the structure [19,59–61].

The temporary qualitative behavior shows that inside the customized structure a greater energy level is generated by the hours of the day, which is translated in higher temperatures between the 7 and the 15 h. This temperature increase is related to higher values of temperature and solar radiation in the external environment of the structure [62,63]. This, together with the structure's own thermal gain, favors this type of thermal behavior that is very characteristic of the protected agricultural structures used in the tropical region of Latin America and the Caribbean [15].

On the other hand, it is also characteristic that as one advances between the afternoon and the night (16 h onwards), a cooling of the structure is generated due to the progressive decrease of the temperature of the exterior air and the level of solar radiation. Under these

conditions the thermal gain of the structure is fully supported by the heat input from the ground to the interior environment of the structure [5,27].

This thermal gain in the internal environment of the structure, with respect to the external environment, can be maintained depending on the level of thermal insulation that the structure has [24]. In this case this level is low since most of the structure is covered by a porous material that allows the exchange of air from the exterior and interior environments, therefore, the temperatures of these two environments tend to equalize quickly (Figure 7).

In quantitative terms, the temperature behavior was analyzed by calculating the average temperature (T) and the thermal differential ($\Delta_T$) between the indoor and outdoor environments for each of the cases simulated on an hourly scale. The standard deviation (SD) value was also calculated to observe numerically the spatial homogeneity in the temperature distribution (Table 5).

**Table 5.** Numerical parameters obtained for temperature distribution.

| Hour | T [°C] | $\Delta_T$ [°C] | Hour | T [°C] | $\Delta_T$ [°C] |
|---|---|---|---|---|---|
| Hour 06 | 21.6 ± 0.11 | 0.31 ± 0.11 | Hour 12 | 33.6 ± 2.22 | 3.88 ± 2.22 |
| Hour 07 | 22.2 ± 0.15 | 0.48 ± 0.15 | Hour 13 | 34.3 ± 2.25 | 3.89 ± 2.25 |
| Hour 08 | 24.9 ± 0.67 | 1.39 ± 0.67 | Hour 14 | 34.8 ± 2.33 | 4.02 ± 2.33 |
| Hour 09 | 28.1 ± 1.52 | 2.54 ± 1.52 | Hour 15 | 34.2 ± 2.20 | 3.38 ± 2.20 |
| Hour 10 | 29.7 ± 1.50 | 2.45 ± 1.50 | Hour 16 | 32.4 ± 1.27 | 2.02 ± 1.27 |
| Hour 11 | 31.5 ± 1.59 | 2.87 ± 1.59 | Hour 17 | 30.3 ± 0.59 | 0.92 ± 0.59 |

The T values ranged from a minimum value of 21.6 ± 0.11 °C for hour 06 to a maximum of 34.8 ± 2.33 °C for hour 14, values that also present a differentiated behavior in terms of homogeneity since for hour 06 the SD was 0.11 °C, while for hour 14 this SD presents a value higher than 2.3 °C. Therefore, the above allows us to identify that for hour 06 the behavior of the microclimate inside the structure is more homogeneous than for hour 14. It should be mentioned that this type of heterogeneous microclimate behavior should be analyzed in agronomic and physiological terms, since, in some species, processes such as transpiration, photosynthesis and nutrient absorption can be affected, which would surely lead to obtaining non-homogeneous productions both in quantity and quality [63,64].

In terms of the average thermal behavior, it should be mentioned that the structure at no point in the evaluated time scale exceeds the value of 35 °C. This temperature value is the maximum recommended to guarantee the adequate growth and development of a high percentage of vegetable species cultivated under protected agricultural systems [65,66]. It is also important to mention that the thermal gradient values were not higher than 4.02 °C. Value that is lower than the $\Delta_T$ reported in other studies where the structure was covered with insect-proof porous mesh and where these $\Delta_T$ are higher than 8 °C [5,45].

To finalize the numerical analysis of the behavior of the temperature in the interior of the structure (T), a linear equation was constructed by relating the values of temperature, solar radiation and wind speed in the exterior environment of the structure for each of the hours evaluated. The data were analyzed by means of a multiple regression analysis to obtain Equation (10).

$$Y = 1.46 + 0.940X_1 + 0.004X_2 + 0.140X_3 \tag{10}$$

where $Y$ represent the value of the temperature inside the structure (°C), $X_1$ is the outside temperature value (°C), $X_2$ is the value of solar radiation (W m$^{-2}$) and $X_3$ is the value of the external wind velocity (m s$^{-1}$). This equation presented a multiple correlation coefficient (r) with a value of 0.997, a determination coefficient (r$^2$) with a value of 0.995 and a r$^2$ adjusted with a value of 0.94, values that allow us to conclude that there is a positive correlation between the data analyzed and that allow us to reaffirm that the behavior of the temperature inside a protected agricultural structure is highly dependent on the

conditions of temperature, radiation and wind speed in the external environment of the structure [67,68].

It is also important to mention that the CFD model validated in this research will become a design and optimization tool that will allow future research to include new design recommendations for the current structure. For example, it will be possible to analyze structural changes such as increasing the height of the structure or evaluating another type of insect-proof porous mesh with a different degree of porosity, include some type of evaporative cooling system or forced ventilation, all strategies aimed at reducing the thermal gradients inside the structure and achieving a higher degree of thermal homogeneity inside the structure. The analysis through numerical simulation will allow the selection of the alternative that really generates a positive impact on the microclimate behavior of the structure. This alternative could be implemented in the structure at full scale and possibly help to improve the technical sustainability of horticultural production in hot climate regions.

## 4. Conclusions

The numerical CFD model implemented within this research, proved to be a tool with a highly satisfactory prediction. Therefore, its use to determine the characteristics of flow patterns and their effect on the thermal distribution within a new protected agricultural structure established in the Dominican Republic is appropriate.

The airflow patterns inside the structure exhibited a reduction in velocity relative to the wind speed in the outside environment. Average speed reductions ranged from 29.7% to 81.7% for the lowest and highest reduction scenarios. These air flow reductions inside the structure are highly influenced by the presence of the porous insect-proof screen located in the ventilation areas, which generates a loss of inertial momentum of the air flow that results in a reduction of air velocity.

The thermal distribution and the magnitude of the average temperature values within the structure were shown to have a direct relationship with the air flow patterns, the level of radiation and the temperature of the outside environment presented in each scenario evaluated. Likewise, the average temperature values inside the structure did not exceed 35 °C, a limiting value for agricultural production in roof structures.

**Author Contributions:** Conceptualization, E.V.; methodology, E.V. and A.R.; software, E.V.; validation, E.V. and A.R.; investigation, E.V. and A.R.; writing—original draft preparation, E.V. and A.R.; writing—review and editing, E.V. and A.R. Both authors have read and agreed to the published version of the manuscript.

**Funding:** The research was funded by The Regional Fund of Agricultural Research and Technological Development (FONTAGRO) as part of the project "Innovations for horticulture in protected environments in tropical zones: an option for sustainable intensification of family farming in the context of climate change in LAC". The opinions expressed in this publication are solely those of the authors and do not necessarily reflect the views of FONTAGRO, its Board of Directors, the Bank, its sponsoring institutions, or the countries it represents.

**Institutional Review Board Statement:** Not applicable.

**Informed Consent Statement:** Not applicable.

**Data Availability Statement:** Not applicable.

**Acknowledgments:** The authors wish to thank Corporación Colombiana de Investigación Agropecuaria (AGROSAVIA) and Instituto Dominicano de Investigaciones Agropecuarias y Forestales (IDIAF) for their technical and administrative support in this study. Additionally, special thanks to the technical team that supported the construction of the structure and those in charge of climate monitoring.

**Conflicts of Interest:** The authors declare no conflict of interest.

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
