# Peer review of "Analysis of the Thermal Behavior of a New Structure of Protected Agriculture Established in a Region of Tropical Climate Conditions"

_fluids, doi:10.3390/fluids6060223_

Round 1

Reviewer 1 Report

From reviewer humble opinion, it is not clear what is the interest for readers about this manuscript. It is not clear authors’ science contribution with that analysis. What is the real applicability? What is the objective of this publication?

  • Section 2.2 should be re-written according to manuscript analysis rather than general description that can be found in general bibliography
  • Avoid Spanish words in Table 2
  • What design recommendations can be proposed based on CFD simulations?

Author Response

Dear reviewer.
Thank you for your valuable suggestions, these contribute to improve the manuscript, therefore, they have been included in the new version of the manuscript. Below you will find the answers to them.

From reviewer humble opinion, it is not clear what is the interest for readers about this manuscript. It is not clear authors’ science contribution with that analysis. What is the real applicability? What is the objective of this publication?

Reply. The authors, following the reviewer's suggestion, have included the following paragraph to answer these concerns both to the reviewer and to the reader of the article. Lines 67 to 73.

“The main objectives of this research were (i) to evaluate and validate a three-dimensional CFD numerical simulation model applicable to a naturally ventilated protected agriculture structure and (ii) to evaluate through numerical simulation using the validated model, the aerodynamic behavior of the air flow patterns and the spatial distribution of the temperature inside a new protected agriculture structure built in a low latitude region in the Dominican Republic. The evaluation and validation of the numerical model was carried out for the daytime hours between 7:00 am and 5:00 pm.”

Section 2.2 should be re-written according to manuscript analysis rather than general description that can be found in general bibliography.

Reply. The methodological section has been rewritten following the reviewer's suggestion. It can be reviewed in the new version of the manuscript between lines 103 to 120.

Avoid Spanish words in Table 2.

Reply. The word has been modified to the English language.

What design recommendations can be proposed based on CFD simulations?

Reply. Following the reviewer's suggestion, the following text has been included in the new version of the article, between lines 417 to 428.

It is also important to mention that the CFD model validated in this research will be-come a design and optimization tool that will allow future research to include new design recommendations for the current structure. For example, it will be possible to analyze structural changes such as increasing the height of the structure or evaluating another type of insect-proof porous mesh with a different degree of porosity, include some type of evaporative cooling system or forced ventilation, all strategies aimed at reducing the thermal gradients inside the structure and achieving a higher degree of thermal homogeneity inside the structure. The analysis through numerical simulation will allow the selection of the alternative that really generates a positive impact on the microclimate behavior of the structure. This alternative could be implemented in the structure at full scale and possibly help to improve the technical sustainability of horticultural production in hot climate regions.”

Reviewer 2 Report

1.Protected agricultural structures are commonly set in many countries and regions to defend crops from natural damage, however, it's rarely to be discussed the parameters like temperature, airflow velocity inside the structure. The paper made the validation from the affection of radiation and outside temperature to inside temperature and provide an effective method to evaluate aerodynamic and thermal behavior to find the solution for high temperature of the structure.

2.Line 208, the row head of Table 2, "Air temperatura" should be modified as "Air temperature".

3.The relationship between inside temperature and outside radiation and temperature was validated with positive trends, it's suggested if the authors can present the  relationship with simplified equations that would provide many agricultural sectors a convenient method to calculate the inside temperature of the agricultural structure in advance.

4.The affection of porous insect screens in airflow  was described in abstract, however it didn't to be mentioned in the conclusion. It's recommended to modify the conclusion.

Author Response

Dear reviewer.
Thank you for your valuable suggestions, these contribute to improve the manuscript, therefore, they have been included in the new version of the manuscript. Below you will find the answers to them.

Line 208, the row head of Table 2, "Air temperatura" should be modified as "Air temperature".

Reply. The word has been modified.

The relationship between inside temperature and outside radiation and temperature was validated with positive trends, it's suggested if the authors can present the relationship with simplified equations that would provide many agricultural sectors a convenient method to calculate the inside temperature of the agricultural structure in advance.

Reply. Following the reviewer's suggestion, the following text has been included in the new version of the article, between lines 401 to 416.

“To finalize the numerical analysis of the behavior of the temperature in the interior of the structure (T), a linear equation was constructed by relating the values of temperature, solar radiation, and wind speed in the exterior environment of the structure for each of the hours evaluated. The data were analyzed by means of a multiple regression analysis to obtain equation 10.

(10)

Where  represent the value of the temperature inside the structure (°C),  is the outside temperature value (°C),  is the value of solar radiation (Wm-2) y  is the value of the external wind velocity (ms-1). This equation presented a multiple correlation coefficient (r) with a value of 0.997, a determination coefficient (r2) with a value of 0.995 and a r2 adjusted with a value of 0.94, values that allow us to conclude that there is a positive correlation between the data analyzed and that allow us to reaffirm that the behavior of the temperature inside a protected agricultural structure is highly dependent on the conditions of temperature, radiation and wind speed in the external environment of the structure [67,68].”

The affection of porous insect screens in airflow  was described in abstract, however it didn't to be mentioned in the conclusion. It's recommended to modify the conclusion.

Reply. Following the reviewer's suggestion, the conclusion has been modified as follows.

The airflow patterns inside the structure exhibited a reduction in velocity relative to the wind speed in the outside environment. Average speed reductions ranged from 29.7% to 81.7% for the lowest and highest reduction scenarios. These air flow reductions inside the structure are highly influenced by the presence of the porous insect-proof screen located in the ventilation areas, which generates a loss of inertial momentum of the air flow that results in a reduction of air velocity.”

Grettings

The authors.

Round 2

Reviewer 1 Report

Authors replied appropiately to reviewer's comments and suggestions